# 1,6-hexanediol rapidly immobilizes and condenses chromatin in living human cells

Yuji Itoh[1], Shiori Iida[1,2], Sachiko Tamura[1], Ryosuke Nagashima[1,2], Kentaro Shiraki[3], Tatsuhiko Goto[4,5], Kayo Hibino[1,2], Satoru Ide[1,2], Kazuhiro Maeshima[1,2]

Liquid droplets formed inside the cell by liquid–liquid phase separation maintain membrane-less condensates/bodies (or compartments). These droplets are important for concentrating certain molecules and facilitating spatiotemporal regulation of cellular functions. 1,6-hexanediol (1,6-HD), an aliphatic alcohol, inhibits weak hydrophobic protein–protein/protein-RNA interactions required for the droplet formation (droplet melting activity) and is used here to elucidate the formation process of cytoplasmic/nuclear condensates/bodies. However, the effect of 1,6-HD on chromatin in living cells remains unclear. We found that 1,6-HD drastically suppresses chromatin motion and hyper-condenses chromatin in human cells by using live-cell single-nucleosome imaging, which detects changes in the state of chromatin. These effects were enhanced in a dose-dependent manner. Chromatin was "frozen" by 5%, or higher, concentrations of 1,6-HD. 1,6-HD greatly facilitated cation-dependent chromatin condensation in vitro. This 1,6-HD action is distinct from its melting activity of liquid droplets. Alcohols, such as 1,6-HD, appear to remove water molecules around chromatin and locally condense chromatin. Therefore, liquid droplet results obtained using 1,6-HD should be carefully interpreted or reconsidered when these droplets are associated with chromatin.

## Introduction

Some macromolecules self-organize into liquid droplets by a process termed liquid–liquid phase separation (LLPS), which allows specific molecules to be concentrated without a membrane, whereas others are excluded (Hyman et al, 2014; Banani et al, 2017; Shin & Brangwynne, 2017). Cells organize liquid droplet-like condensates/bodies (or compartments), contributing to the spatial and temporal regulation of complex biochemical reactions. However, whether all of these dynamic biomolecular condensates/bodies form by LLPS or some form by another process remains unclear and the subject of debate for many cell biologists (McSwiggen et al, 2019).

LLPS is driven by weak and multivalent interactions between proteins and nucleic acids (Banani et al, 2017; Shin & Brangwynne, 2017). In many cases, proteins in liquid droplet-like condensates/bodies have intrinsically disordered regions that lack stable folding and often contain stretches of low sequence complexity (Kato et al, 2012; Elbaum-Garfinkle et al, 2015; Nott et al, 2015; Murray et al, 2017). In principle, intrinsically disordered regions mediate multiple weak and transient reversible interactions, unlike the formation of subcellular aggregates. The molecules inside liquid droplet-like condensates/bodies are highly mobile and can transition in and out of the condensates/bodies (McSwiggen et al, 2019; Taylor et al, 2019).

The aliphatic alcohol, 1,6-hexanediol (1,6-HD) (Fig 1A), has been widely used to study the formation process of the membrane-less cytoplasmic/nuclear condensates/bodies, presumably formed by LLPS (Kroschwald et al, 2017). 1,6-HD inhibits weak hydrophobic protein–protein or protein-RNA interactions required for the formation of liquid droplet-like condensates/bodies (droplet melting activity) (Lin et al, 2016). 1,6-HD was originally noticed for its ability to disrupt FG repeat interactions between nucleoporins in the nuclear pore complex (Ribbeck & Gorlich, 2002; Patel et al, 2007) and interactions between RNA-binding proteins in RNA-protein (RNP) granules (Updike et al, 2011; Kroschwald et al, 2015) in vitro. More recently, 1,6-HD was used to disrupt nuclear condensates/bodies associated with chromatin, which are thought to be formed by LLPS (Cho et al, 2018; Chong et al, 2018; Lu et al, 2018; 2020; Sabari et al, 2018; Yamazaki et al, 2018; Ding et al, 2019; Guo et al, 2019, Kilic et al, 2019; Nair et al, 2019; Han et al, 2020; Crump et al, 2021). Furthermore, 1,6-HD has been used to examine liquid droplet formation of chromatin (Strom et al, 2017; Ulianov et al, 2020 Preprint) or protein/chromatin complexes (Ryu et al, 2020 Preprint; Crump et al, 2021). However, although 1,6-HD is widely used to study protein/RNA condensates/bodies, some reports have pointed out significant limitations and caveats to its use in the context of biomolecular condensates/bodies (Alberti et al., 2019; Kroschwald et al., 2017; Lin et al., 2016; McSwiggen et al., 2019). Indeed, the cellular effects of 1,6-HD, especially its effects on chromatin in living cells, remain unclear.

We investigated how 1,6-HD could influence cellular chromatin behavior in human cells using live-cell single-nucleosome imaging and tracking (Hihara et al., 2012; Lerner et al., 2020; Nagashima et al., 2019;

[1]Genome Dynamics Laboratory, National Institute of Genetics, Mishima, Japan    [2]Department of Genetics, School of Life Science, SOKENDAI, Mishima, Japan    [3]Faculty of Pure and Applied Sciences, University of Tsukuba, Tsukuba, Japan    [4]Research Center for Global Agromedicine, Obihiro University of Agriculture and Veterinary Medicine, Obihiro, Japan    [5]Department of Life and Food Sciences, Obihiro University of Agriculture and Veterinary Medicine, Obihiro, Japan

Correspondence: kmaeshim@nig.ac.jp

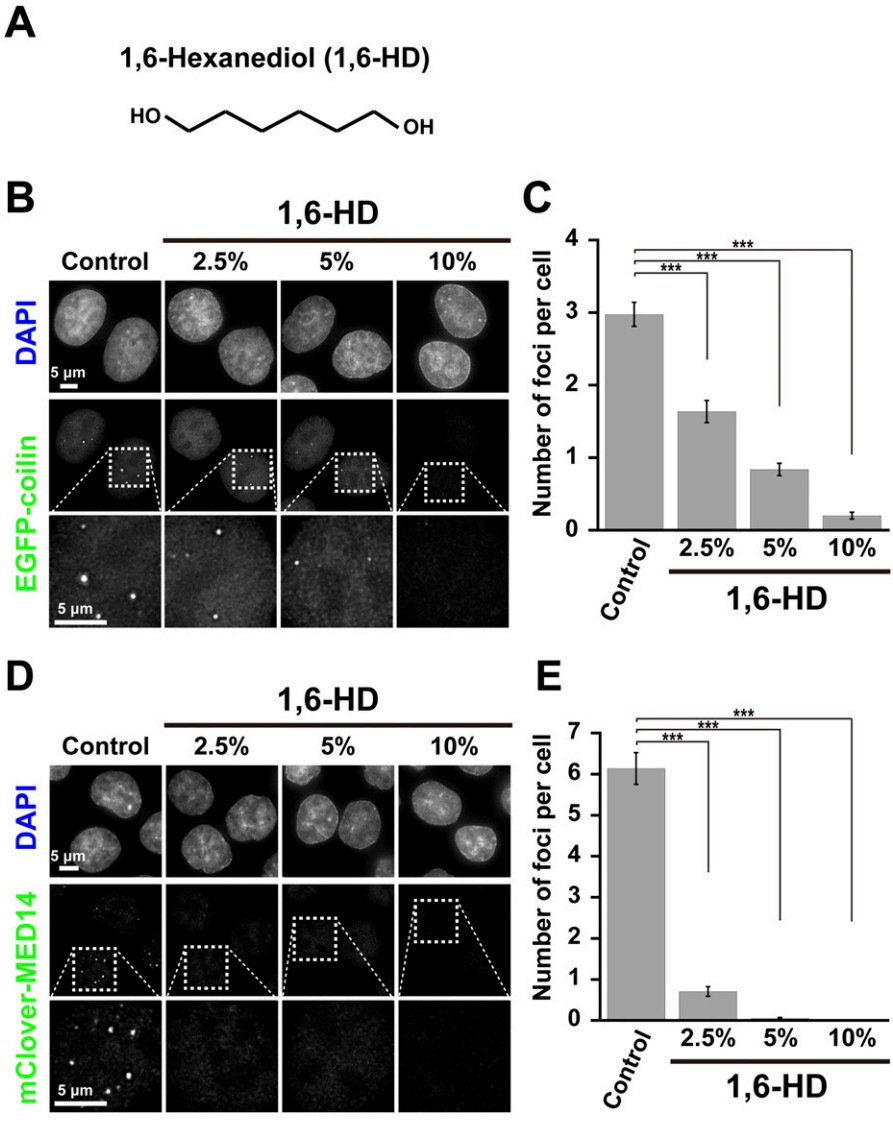

**Figure 1. 1,6-hexanediol (1,6-HD) dissolves nuclear droplets/bodies.**
**(A)** Chemical structure of 1,6-HD. **(B)** Effects of 2.5%, 5%, and 10% 1,6-HD treatments on Cajal bodies labeled with EGFP-coilin in HeLa cells. First row, DNA staining with DAPI; second-row, fluorescent images of EGFP-coilin; third row, magnified images of the boxed regions in the second-row images. **(C)** The quantification of the number of foci per cell is shown as a bar graph. Data are mean ± SEM. The mean number of foci per cell: 2.98 ($n$ = 85 cells) in control; 1.64 ($n$ = 80 cells) in 2.5%; 0.84 ($n$ = 92 cells) in 5%; 0.20 ($n$ = 90 cells) in 10% 1,6-HD. ***$P$ < 0.0001 by the Welch's $t$ test for control versus 2.5% ($P$ = 1.47 × 10$^{-8}$), for control versus 5% ($P$ = 1.31 × 10$^{-21}$), and for control versus 10% ($P$ = 1.31 × 10$^{-29}$). **(D)** Effects of 2.5%, 5%, and 10% 1,6-HD treatments on transcription foci/condensates fluorescently labeled by mClover-MED14 in HCT116 cells. First row, DNA staining with DAPI; second-row fluorescent images of mClover-MED14; third row, magnified images of the boxed regions in the second-row images. **(E)** The quantification of the number of foci per cell is shown as a bar graph. Data are mean ± SEM. The mean number of foci per cell: 6.14 ($n$ = 87 cells) in control; 0.71 ($n$ = 99 cells) in 2.5%; 0.05 ($n$ = 95 cells) in 5%; 0.00 ($n$ = 30 cells) in 10% 1,6-HD. ***$P$ < 0.0001 by the Welch's $t$ test for control versus 2.5% ($P$ = 2.14 × 10$^{-24}$), for control versus 5% ($P$ = 3.46 × 10$^{-27}$), and for control versus 10% ($P$ = 2.15 × 10$^{-27}$).

Nozaki et al, 2017), which can sensitively detect change(s) in chromatin state. Although we confirmed several published results that 1,6-HD treatment disrupted nuclear condensates/bodies (Cho et al., 2018; Lin et al., 2016), we found that 1,6-HD drastically and globally suppressed chromatin motion and hyper-condensed chromatin in live HeLa cells. Similar suppression effects were observed in several other human cell lines and these effects were enhanced in a dose-dependent manner. Chromatin was "frozen" when live cells were treated with 5% or higher 1,6-HD for 5 min. Careful consideration is thus needed to interpret all the results of cell biological experiments performed with 1,6-HD treatment.

## Results

### 1,6-hexanediol treatment disrupts nuclear droplets/condensates

Nuclear condensates/bodies are disrupted by 1,6-hexanediol (1,6-HD) (Fig 1A) (Cho et al., 2018; Lin et al., 2016; Sabari et al., 2018;

Yamazaki et al, 2018). We confirmed these findings by examining Cajal bodies in HeLa cells labeled with EGFP-coilin (Fig 1B). Cajal bodies are nuclear droplets enriched in proteins and RNAs, presumably formed by LLPS. Nuclear foci were observed (Fig 1B), but these foci gradually disappeared with increasing concentrations of 1,6-HD (Fig 1C). 1,6-HD is thought to dissolve Cajal bodies through its droplet melting activity (Lin et al, 2016). Furthermore, transcription condensates/bodies labeled by mClover-MED14 in HCT116 cells were examined (Fig 1D). Again, the number of foci decreased with increasing 1,6-HD treatment (Fig 1E). A similar result was also obtained in transcription condensates/bodies labeled by RNA polymerase II-mClover (mClover-RPB1) in DLD-1 cells (Fig S1) (Nagashima et al, 2019). Our work and others (Cho et al., 2018; Lin et al., 2016; Sabari et al., 2018) demonstrate that 1,6-HD treatments disrupt nuclear condensates/bodies, possibly by its reported droplet melting activity.

### 1,6-HD rapidly suppresses chromatin motion in living human cells

To investigate how 1,6-HD treatment affects chromatin behavior in living cells, we performed single-nucleosome imaging and tracking

(Hihara et al., 2012; Lerner et al., 2020; Nagashima et al., 2019; Nozaki et al., 2017) by oblique illumination microscopy (Fig 2A). This imaging illuminates a thin area within a single nucleus to improve the level of background noise observed (Tokunaga et al, 2008). Our technique sensitively and accurately measures local chromatin dynamics in a whole nucleus and provides new information on how chromatin organizes in living cells. Histone H2B tagged with HaloTag (H2B-Halo) was stably expressed in HeLa cells (Fig S2A). H2B-Halo can be specifically labeled with HaloTag ligand tetramethylrhodamine (TMR) for live-cell imaging (Fig S2B).

Cells were treated with very low concentrations of TMR to obtain sparse labeling (Fig 2B). We recorded the TMR-nucleosome dots (left, Fig 2C) at 50 ms/frame (~100 frames, 5 s total) (Video 1). The dots showed a single-step photobleaching profile (right, Fig 2C), which suggested that each dot represents a single H2B-Halo-TMR molecule in a single nucleosome. The individual dots were fitted with a 2D Gaussian function to estimate the precise position of the nucleosome (Betzig et al, 2006; Rust et al, 2006; Selvin et al, 2007) and were tracked using u-track software (Fig 2D) (Jaqaman et al, 2008) (the position determination accuracy is 15.55 nm). Notably, we tracked only the signals of TMR-labeled H2B-Halo in the nucleosomes (Fig 2C) because free H2B-Halo moved too fast to detect as

dots and track under our imaging conditions. From the nucleosome tracking data, we calculated mean square displacement (MSD), which shows the spatial extent of motion in a certain time period (Dion & Gasser, 2013). The plots of calculated MSD appeared to be sub-diffusive (Fig 2E). Chemical fixation of the cells with formaldehyde (FA) or methanol (MeOH) almost completely immobilized TMR-labeled nucleosomes (Fig 2E), indicating that most of the observed movement was derived from real nucleosome movements in living cells.

The cells were then treated with increasing concentrations of 1,6-HD (0%, 2.5%, 5% and 10%) for 5 min before quantitating movement. The nucleosome motion (MSD) significantly reduced in a dose-dependent manner (Fig 2F and Video 2). Surprisingly, the MSD values obtained using 10% 1,6-HD were similar to those of MeOH-fixed cells (Fig 2E).

### Higher concentrations of 1,6-HD "freeze" chromatin in living human cells

1,6-HD treated cells were extensively washed and chromatin movements were remeasured to determine if the motion suppression effects were reversible or not. The washing step did not

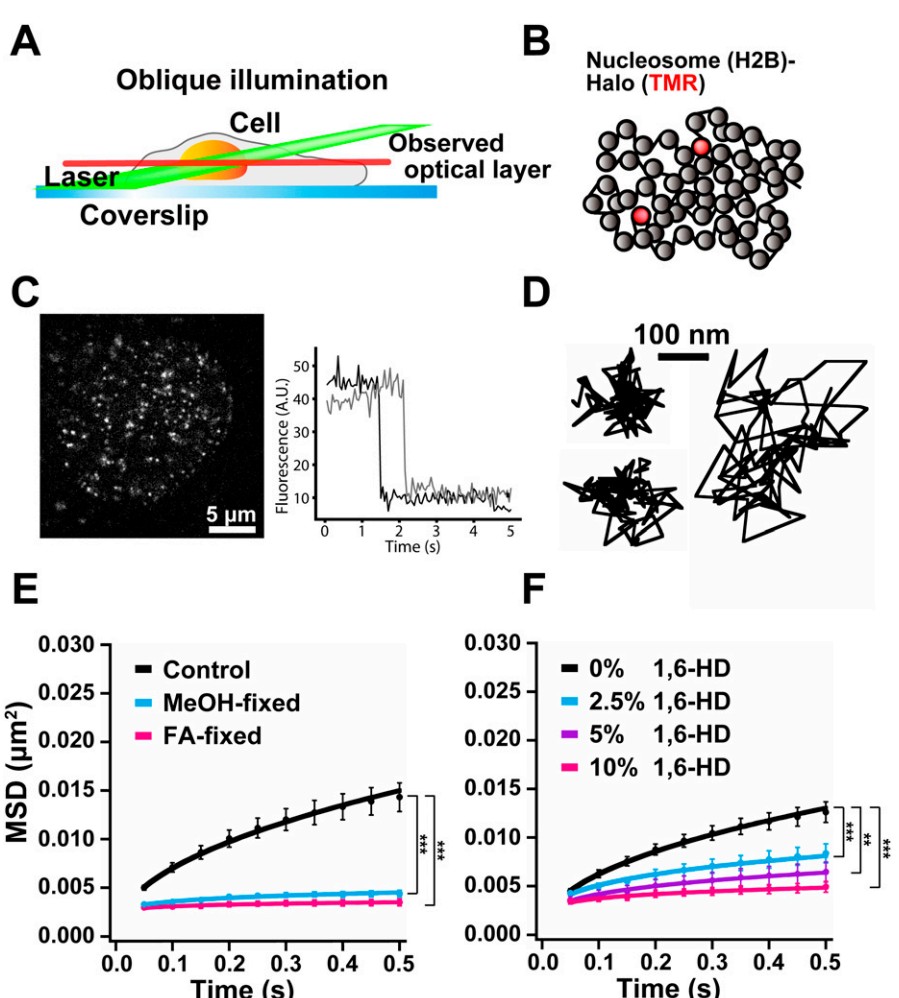

**Figure 2. Single-nucleosome live-cell imaging and the effect of 1,6-HD on nucleosome motion.**
**(A)** Scheme of oblique illumination microscopy. The angled illumination laser (green) can excite fluorescent molecules within a thin optical layer (red) of the nucleus (orange) and reduce background noise. **(B)** A small fraction of H2B-Halo was fluorescently labeled with a low concentration of TMR-HaloTag ligand (red) to obtain sparse labeling. The labeled nucleosome movements can be tracked. **(left, C)** A single-nucleosome (H2B-Halo-TMR) live-cell image of a HeLa nucleus after background subtraction. **(right, C)** Single-step photobleaching of two representative nucleosome (H2B-Halo-TMR) dots. The vertical axis represents the fluorescence intensity of individual TMR dots. The horizontal axis is the tracking time series. The fluorescent intensity of each dot was ~50, and in the single-step photobleaching profile, the intensity dropped to around 10, suggesting that each dot represents a single H2B-Halo-TMR molecule in a single nucleosome. **(D)** Three representative trajectories of the tracked single nucleosomes in HeLa cells. **(E)** Mean square displacement plots (± SD among cells) of single nucleosomes in interphase HeLa cells (Control, black), FA-fixed (pink), and cold methanol-fixed (light blue) HeLa cells. For each condition, $n = 10$ cells. The average numbers of nucleosome trajectories used per cell, 1,300–1,800. ***$P < 0.0001$ for control versus FA-fixed cells ($P = 1.1 \times 10^{-5}$), and for control versus MeOH-fixed cells ($P = 1.1 \times 10^{-5}$). **(F)** Mean square displacement plots (±SD among cells) of nucleosomes in HeLa cells treated with 2.5% (light blue), 5% (purple), or 10% (pink) of 1,6-HD for 5–30 min. For each condition, $n = 8$–10 cells. The average numbers of nucleosome trajectories used per cell, 800–1,800. **$P < 0.01$ for untreated control versus 5% ($P = 1.6 \times 10^{-4}$). ***$P < 0.0001$ for untreated control versus 2.5% ($P = 4.6 \times 10^{-5}$), and for untreated control versus 10% ($P = 4.6 \times 10^{-5}$). Statistical significance in this figure was determined by the Kolmogorov–Smirnov test.

affect chromatin motion in untreated cells (Fig S2C). MSD levels from cells treated with 2.5% 1,6-HD were comparable to untreated cells around 90 min after washing (Fig 3A), suggesting that the effects induced by low levels of 1,6-HD are reversible. However, in treatments with 5% or 10% of 1,6-HD, the reduced MSD values did not change 90 min after washing (Fig 3B and C). These results indicate that a high concentration of 1,6-HD "froze" chromatin and affected chromatin to be similar to that observed in MeOH-fixed cells (Fig 2E).

Cell viability remained comparably high after cells were treated for 30 min with 2.5% or 5% 1,6-HD. However, the viability decreased to about 2% in cells following a 30-min treatment with 10% 1,6-HD (Table 1). These results suggest that our observation of chromatin

"freezing" by 1,6-HD treatment was not a direct consequence of cell death, whereas the treatment has considerable cell toxicity.

## 2,5-hexanediol also suppresses chromatin motion in living human cells

Another aliphatic alcohol, 2,5-hexanediol (2,5-HD) is structurally similar to 1,6-HD (Fig S2D) but has much less melting activity of droplets formed by LLPS (Lin et al, 2016). Therefore, we examined the effect of 2,5-HD on chromatin motion to see if this activity would correlate to its droplet melting activity. The suppression effects on chromatin motion by 2,5-HD at 2.5%, 5%, and 10% were comparable to those by 1,6-HD (Fig 3D). In particular, no significant differences

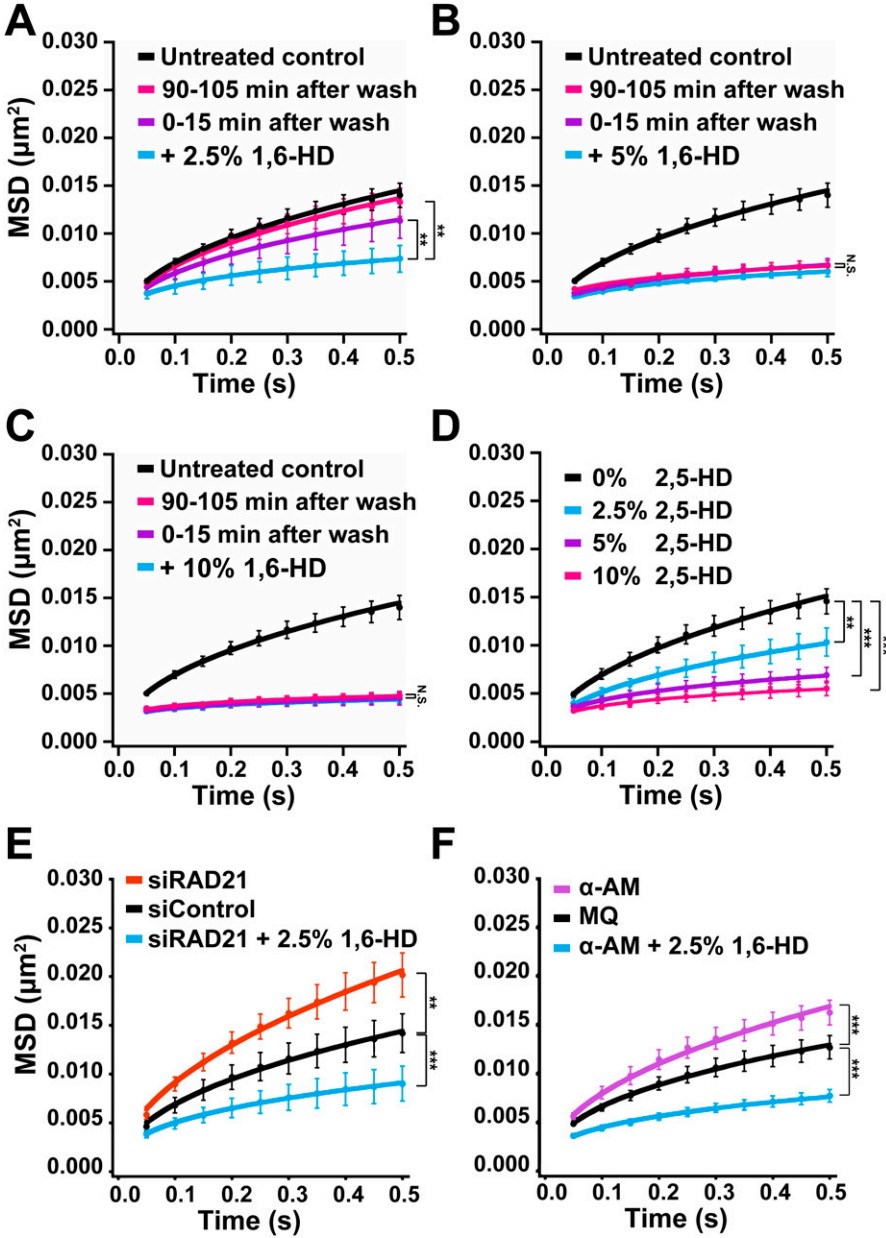

**Figure 3. Effects of 1,6-HD and 2,5-HD on chromatin motion with various conditions in live cells.**
**(A)** Mean square displacement (MSD) plots (±SD among cells) of nucleosomes in HeLa cells with indicated conditions: The cells treated with 2.5% 1,6-HD for 30 min (light blue); 0–15 min after washing out 1,6-HD (purple) and 90–105 min after washing out 1,6-HD (pink). For each condition, $n$ = 7–8 cells. The average numbers of nucleosome trajectories used per cell, 1,000–2,000. **$P$ < 0.01 for + 2.5% 1,6-HD versus 0–15 min after wash ($P$ = 0.0082), and for + 2.5% 1,6-HD versus 90–105 min after wash ($P$ = 3.1 × 10$^{-4}$). **(B)** MSD plots (±SD among cells) of nucleosomes in HeLa cells with indicated conditions: cells treated with 5% 1,6-HD for 30 min (light blue); 0–15 min after washing out 1,6-HD (purple) and 90–105 min after washing out 1,6-HD (pink). For each condition, $n$ = 5–7 cells. The average numbers of nucleosome trajectories used per cell, 400–900. "N.S. (statistically no significance)" for + 5% 1,6-HD versus 0–15 min after wash ($P$ = 0.068), and for + 5% 1,6-HD versus 90–105 min after wash ($P$ = 0.36). **(C)** MSD plots (±SD among cells) of nucleosomes in HeLa cells with indicated conditions: cells treated with 10% 1,6-HD for 30 min (light blue); 0–15 min after washing out 1,6-HD (purple) and 90–105 min after washing out 1,6-HD (pink). For each condition, $n$ = 7–10 cells. The average numbers of nucleosome trajectories used per cell, 800–1,000. "N.S." for + 10% 1,6-HD versus 0–15 min after wash ($P$ = 0.63), and for + 10% 1,6-HD versus 90–105 min after wash ($P$ = 0.25). **(D)** Decreased chromatin motion by 2,5-HD. MSD plots (±SD among cells) of nucleosomes in HeLa cells treated with 2.5% (light blue), 5% (purple), and 10% (pink) 2,5-HD for 5–30 min. For each condition, $n$ = 8–10 cells. The average numbers of nucleosome trajectories used per cell, 600–1,900. **$P$ < 0.01 for untreated control versus 2.5% ($P$ = 2.2 × 10$^{-4}$). ***$P$ < 0.0001 for untreated control versus 5% ($P$ = 4.6 × 10$^{-5}$), and for untreated control versus 10% ($P$ = 1.1 × 10$^{-5}$). **(E)** MSD plots (±SD among cells) of nucleosomes in HeLa RAD21-KD cells untreated (orange) or treated with 2.5% 1,6-HD for 5–30 min (light blue). The control is control siRNA cells (black). For each condition, $n$ = 10 cells. The average numbers of nucleosome trajectories used per cell, 500–1,700. **$P$ < 0.01 for siControl versus siRAD21 ($P$ = 2.2 × 10$^{-4}$). ***$P$ < 0.0001 for siControl versus siRAD21 + 2.5% 1,6-HD ($P$ = 1.1 × 10$^{-5}$). **(F)** MSD plots (±SD among cells) of nucleosomes in HeLa cells treated with RNAPII inhibitor, $\alpha$-amanitin ($\alpha$-AM, purple) or $\alpha$-AM and 2.5% 1,6-HD for 5–60 min (light blue). The control is Milli-Q water (MQ, black). For each condition, $n$ = 10 cells. The average numbers of nucleosome trajectories used per cell, 500–2,300. ***$P$ < 0.0001 for MQ versus $\alpha$-AM ($P$ = 5.1 × 10$^{-5}$), for MQ versus $\alpha$-AM + 2.5% 1,6-HD ($P$ = 6.4 × 10$^{-7}$). Statistical significance in this figure was determined by the Kolmogorov–Smirnov test.

**Table 1. Cell viability of HeLa cells treated with 1,6-HD.**

| 1,6-HD (%) | 0% | 2.5% | 5% | 10% |
|---|---|---|---|---|
| Cell viability (%) | 97% ± 3% | 97% ± 1% | 96% ± 2% | 2% ± 1% |

Cell viability after treatment of the indicated concentration of 1,6-HD is given as the mean ± standard deviation following treatment with increasing concentrations of 1,6-HD. Experiments were performed in triplicate with ~3 × $10^5$ cells/each experiment.

were observed between 2,5-HD and 1,6-HD at 5% or 10% (Fig 3D) ($P$ = 0.28 by the Kolmogorov–Smirnov test for 5% 2,5-HD versus 5% 1,6-HD [Fig 2F]; $P$ = 0.42 for 10% 2,5-HD versus 10% 1,6-HD [Fig 2F]). This finding suggests that the observed 1,6-HD effect on chromatin motion in living cells is not directly related to the melting activity of liquid droplets or disruptions of weak hydrophobic interactions between proteins/RNAs/DNAs in the droplets, as previously reported for 1,6-HD mechanism of action (Lin et al, 2016).

### 1,6-HD directly alters chromatin, rather than affecting chromatin indirectly through chromatin-bound proteins

We wondered whether 1,6-HD directly influenced chromatin or if chromatin effects were indirectly caused by altering chromatin-bound proteins, as many of these proteins can constrain chromatin (Babokhov et al, 2020). To this end, we knocked down RAD21, part of the cohesin complex (Fig S2E) (Nasmyth & Haering, 2005; Nishiyama, 2019), and examined the effect of 1,6-HD on chromatin motion in cohesin-depleted cells (Fig 3E). Cohesin can encircle chromatin fibers with its ring structure (Nasmyth & Haering, 2005; Nishiyama, 2019) and constrain chromatin motion (Dion et al, 2013; Nozaki et al, 2017). Chromatin motion significantly increased in cells depleted of RAD21 (Figs 3E and S2E), as consistent with the previous report (Nozaki et al, 2017). However, the addition of 2.5% 1,6-HD suppressed the knockdown effect and further lowered the chromatin movement (Fig 3E). Chromatin motion in RAD21-depleted cells treated with 2.5% 1,6-HD was comparable to cells only treated with 2.5% 1,6-HD (Fig 2F), with no statistical significance ($P$ = 0.17 by the Kolmogorov–Smirnov test for 2.5% 1,6-HD [Fig 2F] versus siRAD21 + 2.5% 1,6-HD).

A similar result was obtained when transcription machinery was inhibited. Unexpectedly, transcription machinery like RNA polymerase II (RNAPII) constrains chromatin motion in the cell (Babokhov et al., 2020; Nagashima et al., 2019): Treatment with the RNAPII inhibitor (α-amanitin) reduced the amount of active RNAPII (Fig S2F) and increased chromatin movements (Fig 3F). The chromatin motion with combined treatments of α-amanitin and 1,6-HD drastically decreased (Fig 3F), and was also not significantly different to cells only treated with 2.5% of 1,6-HD (Fig 2F) ($P$ = 0.34 by the Kolmogorov–Smirnov test for 2.5% 1,6-HD [Fig 2F] versus α-AM + 2.5% 1,6-HD). Taken together, these results suggest that 1,6-HD directly acts on chromatin.

### 1,6-HD has similar effects on several human cells

We investigated chromatin motion in three other human cell lines: RPE-1 (Bodnar et al, 1998), HCT116, and DLD-1 to exclude the possibility that the 1,6-HD chromatin effects were unique to HeLa cells.

We performed single-nucleosome imaging and tracking for RPE-1, HCT116, and DLD-1 cells, all of which stably expressed H2B-Halo. Similar to HeLa cells treated with 1,6-HD, suppression of chromatin motion was observed in a dose-dependent manner in RPE-1, HCT116, and DLD-1 cells treated with 1,6-HD (Fig 4A–C). Whereas the potency at each concentration varied at lower doses, treatments of 10% 1,6-HD seemed equivalent among all cell lines tested. Collectively, effects by 1,6-HD on chromatin motion appear to be general, not specific to particular cell types.

### 1,6-HD condenses chromatin structure in live cells

We investigated how 1,6-HD influences chromatin structure/ organization in living cells because motion suppression effects described above should be reflected in structural changes of chromatin when cells are treated with 1,6-HD. For this purpose, we used photoactivated localization microscopy (PALM) (Betzig et al, 2006; Manley et al, 2008; Nozaki et al, 2017) to perform super-resolution live-cell imaging of HeLa cells expressing histone H2B tagged with photoactivatable (PA)-mCherry (Subach et al, 2009). We reconstructed the spatial organization of nucleosomes from the obtained PALM images (Fig 4D).

Cells were treated with 2.5%, 5%, or 10% of 1,6-HD for 5 min before PALM imaging and reconstructing the high-resolution chromatin images (Fig 4D). We found that chromatin seemed more condensed with increasing amounts of 1,6-HD (Fig 4D). L function, L(r), was used to quantitate nucleosome clustering (Fig S3A) (Nozaki et al, 2017). The L-function plot (L(r)-r versus r plot) gives a value of 0 for the random distribution (blue, Fig S3A), and deviation from zero provides an intuitive measure of the size of the cluster and the degree of accumulation (red, Fig S3A) (Nozaki et al, 2017). The L-function plot peak provides good approximations of the size and compaction state of the nucleosome clusters (or chromatin domains). The L-function plots (Fig 4E) suggest that 5% and 10% 1,6-HD both caused chromatin hyper-condensation, whereas cells treated with 2.5% 1,6-HD had a somewhat similar nucleosome clustering to that in untreated control cells (Fig 4E). The hyper-condensation effects of 5% and 10% 1,6-HD could be correlated to their observed immobilization effects on nucleosomes (Fig 3B and C).

### 1,6-HD facilitates chromatin condensation in vitro

We examined the effects of 1,6-HD on $Mg^{2+}$-dependent chromatin condensation in vitro to further investigate how 1,6-HD induces hyper-condensation of chromatin. Chromatin, which is negatively charged and repulses other chromatin in the absence of cations, is neutralized by $Mg^{2+}$ and condensed in a dose-dependent manner (Fig S3B) (Hansen, 2002; Maeshima et al, 2016, 2018). Purified chicken native chromatin was used for our study (Fig S3C and D). We observed condensates (~1 μm in size) when purified chromatin was mixed with 2.5 mM $Mg^{2+}$ and stained with 4′,6-diamidino-2-phenylindole (DAPI) (Fig 5A). To quantitate the chromatin condensation process, we performed a static light scattering assay (Dimitrov et al, 1986) with a titration of $Mg^{2+}$. Dramatic chromatin condensation was observed in the range of 1.5–2 mM $Mg^{2+}$ (Fig 5B). When increasing concentrations of 1,6-HD were added, the scattering plots were shifted to the left (Fig 5B). This shift indicated that

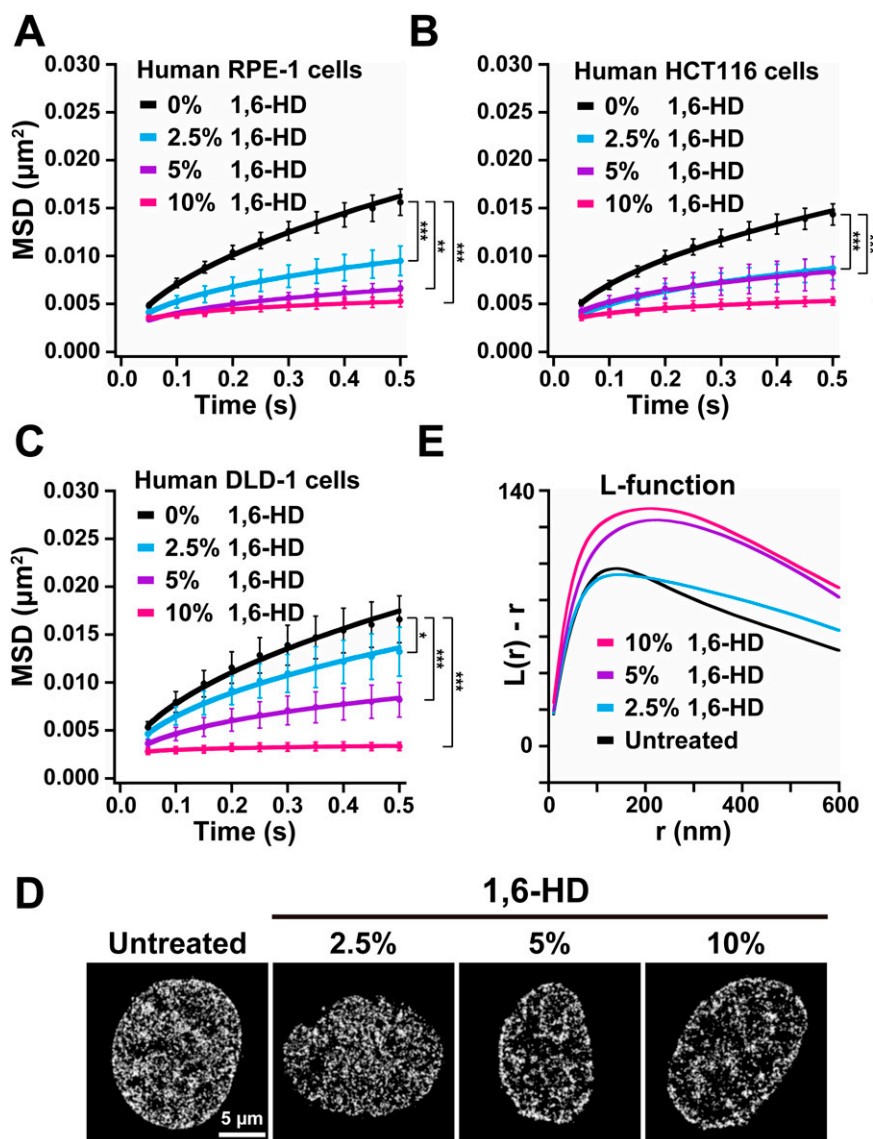

**Figure 4. 1,6-HD effects on chromatin motion in various cells and on chromatin structure and organization revealed at high resolution.**
**(A)** Mean square displacement (MSD) plots (±SD among cells) of nucleosomes in the human RPE-1 cells treated with 2.5% (light blue), 5% (purple), and 10% (pink) 1,6-HD for 5–30 min. For each condition, $n$ = 7–11 cells. The average numbers of nucleosome trajectories used per cell, 500–1,400. **$P$ < 0.01 for untreated control versus 5% ($P$ = 1.0 × 10$^{-4}$). ***$P$ < 0.0001 for untreated control versus 2.5% ($P$ = 2.2 × 10$^{-5}$), and for untreated control versus 10% ($P$ = 5.7 × 10$^{-6}$). **(B)** MSD plots (±SD among cells) of nucleosomes in the human HCT116 cells treated with 2.5% (light blue), 5% (purple), and 10% (pink) 1,6-HD for 5–30 min. For each condition, $n$ = 9–11 cells. The average numbers of nucleosome trajectories used per cell, 700–1,300. ***$P$ < 0.0001 for untreated control versus 2.5% ($P$ = 1.2 × 10$^{-5}$), for untreated control versus 5% ($P$ = 5.7 × 10$^{-6}$), and for untreated control versus 10% ($P$ = 5.7 × 10$^{-6}$). **(C)** MSD plots (±SD among cells) of nucleosomes in the human DLD-1 cells treated with 2.5% (light blue), 5% (purple), and 10% (pink) 1,6-HD for 5–30 min. For each condition, $n$ = 9–10 cells. The average numbers of nucleosome trajectories used per cell, 600–1,500. *$P$ < 0.05 for untreated control versus 2.5% ($P$ = 0.045). ***$P$ < 0.0001 for untreated control versus 5% ($P$ = 2.2 × 10$^{-5}$), and for untreated control versus 10% ($P$ = 2.2 × 10$^{-5}$). **(D)** PALM images of interphase chromatin based on live-cell imaging of H2B-PA-mCherry in HeLa cells. From left to right, shown are a control (untreated) cell, cells treated with 2.5%, 5%, or 10% 1,6-HD for 5 min. The average numbers of nucleosome dots used per cell, 28,000–40,000. **(D, E)** L-function plots of chromatin with the same conditions as in (D). For each condition, $n$ = 10 cells. For L-function plot, see Fig S3. Statistical significance in this figure was determined by the Kolmogorov–Smirnov test.

the addition of 1,6-HD facilitated chromatin condensation at lower concentrations of Mg$^{2+}$. The facilitation effect was greatly enhanced as increasing concentrations of 1,6-HD were used (2.5–10%) (Fig 5B). These results indicate that 1,6-HD directly acts on chromatin and promotes chromatin condensation in vitro, consistent with the previous microscopic observations in live cells (Fig 4D and E).

## Discussion

Our single-nucleosome imaging/tracking revealed that the aliphatic alcohol 1,6-HD, which has been widely used for LLPS studies, can immobilize chromatin motion and hyper-condense chromatin in live cells. Single-molecule imaging is sufficiently sensitive to detect possible change(s) of local chromatin environments when treated in real time in live cells, which other imaging or genomic techniques might not see. Interestingly, another aliphatic alcohol,

2,5-HD, which has a much lower melting activity of droplets formed by LLPS (Lin et al, 2016), had a comparable motion suppression effect to 1,6-HD (Fig 3D). This finding indicates that the observed 1,6-HD "freezing" action on chromatin organization is distinct from its disruption activity of liquid droplets formed by LLPS.

To better understand what may be happening in a cell when it is treated with 1,6-HD, it is useful to discuss the general properties of alcohols. Alcohol concentrations above 40% denature protein structure by strengthening intramolecular hydrogen bonds (Shiraki et al, 1995), whereas a low percentage of alcohol does not affect the protein structure or solubility (Chin et al, 1994). However, these low concentrations of alcohol weaken the hydrophobic interactions between proteins, allowing alcohol to dissolve or melt the protein droplets without protein denaturation. Indeed, the more hydrophobic 1,6-HD is known to dissolve protein droplets better than 2,5-HD in vitro (Lin et al, 2016).

1,6-HD has been identified as a compound that can be used to distinguish between liquid-like and solid-like structures in vitro

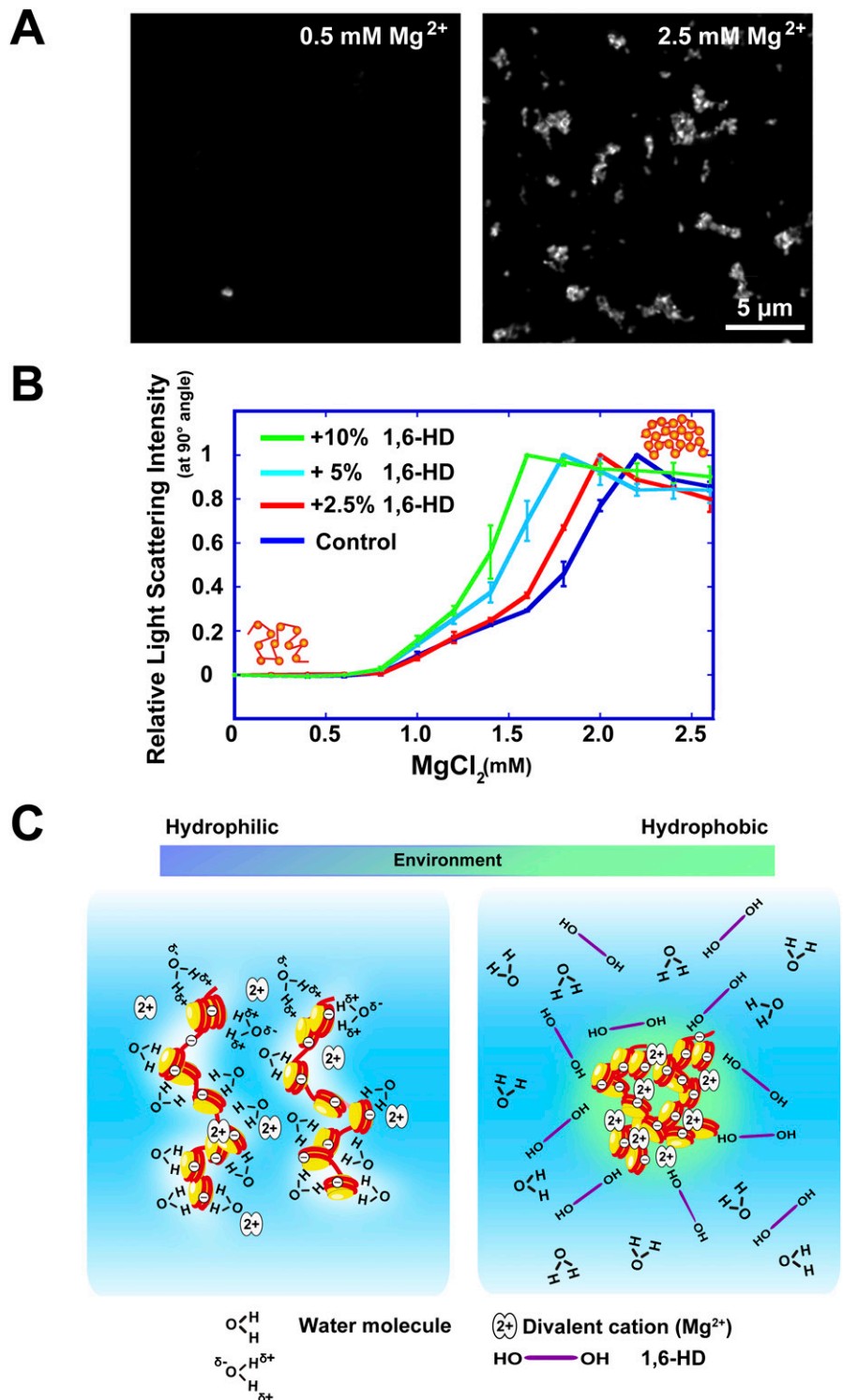

**Figure 5. Enhancement of Mg²⁺-dependent chromatin condensation by 1,6-HD and summary model.**
**(A)** Mg²⁺-dependent chromatin condensates stained with DAPI. Chromatin condensates were formed with 2.5 mM Mg²⁺, but not with 0.5 mM Mg²⁺. **(B)** The effects of 1,6-HD on Mg²⁺-dependent chromatin condensation were examined using static light scattering analysis of purified chromatin. SD is shown by bars ($n$ = 4 experiments). The plots were greatly shifted with the addition of 1,6-HD, indicating enhancement of Mg²⁺-dependent chromatin condensation by 1,6-HD. **(C)** (Left) Chromatin is associated with many water molecules with electrostatic interactions. (Right) Alcohols such as 1,6-HD can remove water molecules around chromatin, and its environment becomes more hydrophobic. This environmental change facilitates the formation of chromatin condensates. Note, this scheme is highly simplified and the molecules shown are not to scale. How 1,6-HD acts on chromatin at molecular level remains unclear.

and in living cells (Kroschwald et al., 2017; Lin et al., 2016). However, it should be noted that this property of 1,6-HD works properly if the liquid droplets are composed of proteins or possibly proteins bound with RNAs. The characteristics of the liquid droplets associated with chromatin should greatly differ from those of protein-only or protein/RNA liquid droplets.

Although the mechanism on how 1,6-HD acts on chromatin remains unclear, we consider that alcohols such as 1,6-HD might remove water molecules around chromatin and locally condense chromatin as condensates (right, Fig 5C) because each nucleosome binds ~3,000 water molecules (Davey et al, 2002) and the surrounding chromatin environment is highly hydrophilic (left, Fig 5C).

This notion reminds us of "ethanol precipitation" to recover purified plasmid or genomic DNAs (Sambrook & Russell, 2001). Indeed, 5% of 1,6-HD "froze" chromatin motion in live cells and the suppressed motion did not recover over 90 min after washing out 1,6-HD (Fig 3B). This situation appears to be similar to that observed with methanol fixation (Fig 2E). Another mechanism of how 1,6-HD acts on chromatin in the cell is also possible, and further investigation is warranted to gain added mechanistic insight into this intriguing issue.

As discussed above, the effect of 1,6-HD on chromatin in living cells is distinct from the melting activity of liquid droplets or disruptions of weak hydrophobic interactions between proteins/RNAs/DNAs in droplets. Dissolving LLPS driven formation of cytoplasmic/nuclear condensates/bodies was previously reported to be the main action of 1,6-HD in biological studies (Lin et al, 2016). Although we agree that the use of 1,6-HD is remarkably effective for simplified in vitro experiments, caution should be used as 1,6-HD treatment can, directly or indirectly, affect various kinds of interactions between DNAs/RNAs/proteins. Thus, careful interpretation of the results obtained from cell biological experiments using 1,6-HD treatment should be done. Our study also suggests that 1,6-HD-sensitivity cannot be evidence for proving that cellular condensates/bodies of protein/DNA complexes, including chromatin, are formed by LLPS. More quantitative analyses of the molecular behavior in condensates/bodies in living cells would be required (McSwiggen et al, 2019; Taylor et al, 2019), such as single-molecule tracking (McSwiggen et al, 2019; Ide et al, 2020).

# Materials and Methods

### Cell lines, DNA construction, and establishment of stable cell lines

HeLa S3 cells (Maeshima et al, 2006) were cultured at 37°C in 5% $CO_2$ in DMEM (D5796-500ML; Sigma-Aldrich) supplemented with 10% FBS (FB-1061/500; Biosera). Human DLD-1 cells (CCL-221; ATCC) expressing mClover-RPB1 and H2B-Halo (Nagashima et al, 2019) and HCT116 cells (CCL-247; ATCC) expressing H2B-Halo (Nagashima et al, 2019) were cultured at 37°C in 5% $CO_2$ in RPMI-1640 medium (R8758-500Ml; Sigma-Aldrich) supplemented with 10% FBS and McCoy's 5A medium (SH30200.01; HyClone) supplemented with 10% FBS, respectively. HCT116 cells expressing mClover-MED14 were kindly gifted by Dr. Masato T Kanemaki at the National Institute of Genetics (Japan) and maintained with the same condition as HCT116 cells expressing H2B-Halo.

The transposon system was used to stably express H2B-Halo in the HeLa S3 cell line. The constructed plasmid pPB-CAG-IB-H2B-HaloTag (Nozaki et al, 2017) was cotransfected with pCMV-hyPBase (provided from Sanger Institute with a materials transfer agreement) to HeLa S3 cells with the Effectene transfection reagent kit (301425; QIAGEN). Transfected cells were then selected with 10 μg/ml blasticidin S (029–18701; Wako).

Lysates from transfected HeLa S3 cells, equivalent to $1 \times 10^5$ cells per well, were subjected to SDS-polyacrylamide gel (12.5%) electrophoresis and transferred to a PVDF membrane (IPVH00010; Millipore) by a semi-dry blotter (BE-320; BIO CRAFT) to confirm H2B-Halo expression. After blocking with 5% skim milk (Morinaga), the membrane bound fractionated cell lysates were probed by the anti-H2B rabbit (1:10,000 dilution; ab1790; Abcam) or anti-HaloTag mouse (1:

1,000; G9211; Promega) antibody, followed by the suitable secondary antibody: anti-rabbit (1:5,000 dilution; 170-6515; Bio-Rad) or anti-mouse (1:5,000 dilution; 170-6516; Bio-Rad) horseradish peroxidase-conjugated goat antibody. Chemiluminescence reactions were used (WBKLS0100; Millipore) and detected by EZ-Capture MG (AE-9300H-CSP; ATTO).

H2B-Halo localization in HeLa S3 cells was determined by treating cells grown on the poly-L-lysine (P1524-500MG; Sigma-Aldrich) coated coverslips (C018001; Matsunami) with 5 nM Hal-oTag TMR Ligand (8251; Promega) overnight at 37°C in 5% $CO_2$. Following processes were performed at room temperature. After washing with PBS, cells were fixed in 1.85% FA (Wako) in PBS for 15 min and then treated with 50 mM glycine in HMK (20 mM Hepes [pH 7.5] with 1 mM $MgCl_2$ and 100 mM KCl) for 5 min and permeabilized with 0.5% Triton X-100 in HMK for 5 min. After washing with HMK for 5 min, the cells were stained with 0.5 μg/ml 4′,6-diamidino-2-phenylindole (DAPI) for 5 min, followed by washing with HMK. Coverslips containing the stained cells were mounted in PPDI (20 mM Hepes [pH 7.4], 1 mM $MgCl_2$, 100 mM KCl, 78% glycerol, and 1 mg/ml paraphenylene diamine [695106-1G; Sigma-Aldrich]) and sealed with a nail polish (T and B; Shiseido).

Z-stack images (every 0.2 μm in the z direction, 20–25 sections in total) of the cells were obtained using DeltaVision Elite microscopy (Applied Precision) with an Olympus PlanApoN 60× objective (NA 1.42) and a sCMOS camera. InsightSSI light (~50 mW) and the four-color standard filter set were also equipped. DeltaVision acquisition software, Softworx, was used to project deconvolved z-stacks to cover the whole nucleus (seven images) because the signals were not distributed homogeneously across all the z-stacks.

Plasmid construction and establishment of HeLa S3 cells stably expressing EGFP-coilin were as follows: To clone full-length coilin, total RNA was isolated from human RPE-1 cells using an RNeasy Mini Kit (74104; QIAGEN) and first-strand cDNA was synthesized using a SuperScript III First-Strand Synthesis System (18080-400; Thermo Fisher Scientific) with oligo (dT). The coding region of coilin was amplified from first-strand cDNA using the following primers: 5′-TCTGGTGGCGGCGGTTCAATGGCAGCTTCCGAGACGGTTAG-3′ and 5′-GCCACTGTGCTGGATTCAGGCAGGTTCTGTACTTGATGTG-3′. The EGFP fragment was amplified from pEGFP-C1-Fibrillarin (#26673; Addgene) (Chen & Huang, 2001) using the following primers: 5′-TGG AATTCTGCAGATGCCACCATGGTGAGCAAGGGCGAGGA-3′ and 5′-CGCCG CCACCAGATCCACCTCCACCAGATCCACCTCCACCCTTGTACAGCTCGTCCATG CCG-3′. The amplified coilin and EGFP fragments were joined together using standard overlapping PCR and inserted into the EcoRV site of a pEF1-FRT plasmid (Maeshima et al, 2010) to obtain pEF1-EGFP-coilin-FRT using In-Fusion (639650; Takara).

HeLa S3 cells stably expressing EGFP-coilin were established using an Flp-In system (K601002; Invitrogen) as previously described (Hihara et al, 2012). pEF1-EGFP-coilin-FRT was transfected into HeLa S3 cells that harbored an FRT site, and transformants were selected using 200 μg/ml hygromycin B (10687010; Invitrogen).

### Imaging and quantification of fluorescent Cajal bodies and transcription condensates

HeLa S3 cells expressing EGFP-coilin, HCT116 cells expressing mClover-MED14 and DLD-1 cells expressing mClover-RPB1 were used. Cells grown on the poly-L-lysine coated coverslips were

treated with 0%, 2.5%, 5%, and 10% (wt/vol) 1,6-hexanediol (1,6-HD) (240117-50G; Sigma-Aldrich) for 5 min. The treated cells were fixed in 2% FA at 37°C for 15 min, permeabilized, stained for DNA, and mounted as described above.

Optical sectioning images were recorded with a 400 nm step size using a DeltaVision microscope (Applied Precision) as described above. Softworx was used to project acquired images over the whole nucleus (usually five images). The projected images were deconvolved and used as source images. Nucleoplasm regions were extracted on the basis of the DNA (DAPI) staining regions.

A median filtered image (radius = 8 pixel) was subtracted from the source image using ImageJ software (NIH) to count the number of fluorescent foci/condensates/bodies. The processed image was smoothed by adding a Gaussian blur ($\sigma$ = 1 pixel). Then, a threshold was applied to count the number of local maxima above background in cells (Cho et al, 2018).

### Single-nucleosome imaging microscopy

Established cell lines were cultured on poly-L-lysine coated glass-based dishes (3,970-035; Iwaki). H2B-Halo molecules were fluorescently labeled with 80 pM HaloTag TMR ligand for 20 min at 37°C in 5% $CO_2$, washed with 1× HBSS (H1387; Sigma-Aldrich) three times, and then incubated in medium without phenol red for more than 30 min before live-cell imaging. HeLa S3 and RPE-1 cells were observed in DMEM (21063-029; Thermo Fisher Scientific), and DLD-1 and HCT116 cells were observed in RPMI-1640 (11835-030; Thermo Fisher Scientific) and McCoy's 5A (1-18F23-1; BioConcept) media, respectively. All of the media were phenol red free and supplemented with 10% FBS.

The live-cell chamber INU-TIZ-F1 (Tokai Hit) and GM-8000 digital gas mixer (Tokai Hit) were used to maintain cell culture conditions (37°C, 5% $CO_2$, and humidity) during microscopy. Single nucleosomes were observed by using an inverted Nikon Eclipse Ti microscope with a 100-mW Sapphire 561-nm laser (Coherent) and sCMOS ORCA-Flash 4.0 camera (Hamamatsu Photonics). Live cells fluorescently labeled with H2B-Halo-TMR or PA-mCherry were excited by the 561-nm laser through an objective lens (100× PlanApo TIRF, NA 1.49; Nikon) and detected at 575–710 nm. An oblique illumination system with the TIRF unit (Nikon) was used to excite fluorescent molecules within a limited thin area in the cell nucleus and reduce background noise. Sequential image frames were acquired using MetaMorph software (Molecular Devices) at a frame rate of 50 ms under continuous illumination.

### Single-nucleosome tracking analysis

Image processing, single-molecule tracking, and single-nucleosome movement analysis were performed as previously described (Nagashima et al., 2019; Nozaki et al., 2017). Sequential images were converted to 8-bit grayscale, and the background noise signals were subtracted with the rolling ball background subtraction (radius, 50 pixels) of ImageJ. The nuclear regions in the images were manually extracted. Following this step, the centroid of each fluorescent dot in each image was determined, and its trajectory was tracked with u-track (MATLAB package; [Jaqaman et al, 2008]). To generate photoactivated localization microscopy (PALM)

images, the individual nucleosome positions were mapped using R software (65 nm/pixel) on the basis of the u-track data, and then a Gaussian blur ($\sigma$ = 1 pixel) was added to obtain smoother rendering using ImageJ. For single-nucleosome movement analysis, the MSD of the fluorescent dots was calculated on the basis of their trajectory using a Python program (Nagashima et al, 2019). The originally calculated MSD was in 2D. To obtain the 3D value, the 2D value was multiplied by 1.5 (4–6 Dt). Statistical analyses of the obtained single-nucleosome MSD between various conditions were performed using R.

### Clustering analyses of nucleosomes in PALM images

The methods for clustering analyses of nucleosomes in PALM images were described previously (Nozaki et al, 2017). Ripley's $K$ function is given by

$$K(r) = \left( \frac{S}{N-1} \right) \left[ \frac{1}{N} \sum_{i=1}^{N} \sum_{i \neq j} \delta(r - r_{i,j}) \right],$$

where $(N-1)/S$ is the average particle density of area $S$, and $N$ is the total number of particles contained in the area. The $\delta$ function is given by

$$\delta(r - r_{i,j}) = \begin{cases} 1, & r_{i,j} \leq r \\ 0, & r_{i,j} > r \end{cases},$$

where $r_{i,j}$ is the distance between $r_i$ and $r_j$.

The $L$ function is given by

$$L(r) = \sqrt{\frac{K(r)}{\pi}} .$$

The area $S$ of the total nuclear region was estimated using the Fiji plugin Trainable Weka Segmentation, and the area of the whole region was measured by Analyze Particles (ImageJ).

### Chemical treatment in single-nucleosome imaging

For chemical fixation, cells grown on poly-L-lysine coated glass-based dishes were incubated in 2% FA (Wako) in 1 × HBSS at 37°C for 15 min or in 100% methanol at –30°C for 15 min and washed with 1 × HBSS. Cells grown in 500 $\mu$l of phenol red free medium supplemented with 10% FBS on poly-L-lysine coated glass-based dishes were treated with pre-warmed 500 $\mu$l solutions of 5%, 10%, or 20% (wt/vol) 1,6-HD or 2,5-hexanediol (2,5-HD) (H11904-10G; Sigma-Aldrich) for 5 min in phenol red–free medium supplemented with 10% FBS to generate hexanediol concentrations of 2.5%, 5%, or 10%. Then single-nucleosome imaging was performed with 1,6-HD or 2,5-HD for up to 30 min.

### Cell viability assay

HeLa S3 cells were treated with 0%, 2.5%, 5%, and 10% (wt/vol) 1,6-HD for 30 min in DMEM supplemented with 10% FBS. After washing with PBS and trypsinization, cells were resuspended in DMEM supplemented with 10% FBS, and were stained for 3 min with 0.1%

trypan blue (29853-34; Nacalai Tesque). The number of viable cells was counted using the TC20 Automated Cell Counter (1450101J1; Bio-Rad).

### RNA interference and α-amanitin (α-AM) treatment

siRNA transfection into HeLa S3 cells grown on poly-L-lysine coated glass-based dishes was performed using Lipofectamine RNAiMAX (13778-075; Invitrogen) according to the manufacturer's instructions. The medium was changed to a fresh medium 16 h after transfection. The transfected cells were used for subsequent studies 48 h after transfection. The siRNA oligonucleotide targeting RAD21 sequence (Sense: 5′-CAGCUUGAAUCAGAGUAGAGUGGAA-3′; Invitrogen) was used. As a control, an oligonucleotide (4390843; Ambion; the sequence is undisclosed) was used. For double treatment with RAD21-KD and 2.5% 1,6-HD, cells were cultured for 48 h after RAD21 siRNA transfection and then treated with 2.5% 1,6-HD as described above.

For transcription inhibition, cells were treated for 4 h with the transcription inhibitor, 100 $\mu$g/ml $\alpha$-AM (A2263-1MG; Sigma-Aldrich). Cells were imaged or chemically fixed in FA after the treatment. For double treatment with $\alpha$-AM and 2.5% 1,6-HD, cells were treated with 1 ml medium containing 100 $\mu$g/ml $\alpha$-AM for 4 h, then an equal volume of medium containing 5% 1,6-HD was added on to the glass-based dish on the microscope just before observation.

### Indirect immunofluorescence

To verify RAD21-depletion and transcription inhibition, immuno-staining was performed as described previously (Hihara et al, 2012), and all processes were performed at room temperature. Cells on the coverslips were fixed and permeabilized as described above. After washing twice with HMK for 5 min, the cells were incubated with 10% normal goat serum (NGS; 143-06561; Wako) in HMK for 30 min. The cells were incubated with diluted primary antibodies: mouse anti-RAD21 (1:1,000 dilution, 05-908; Upstate) or mouse anti–phosphorylated Ser5 of RNA Polymerase II (RNAPII) (1:1,000, RNAPII-Ser5P provided by Dr. H Kimura; clone CMA603 described in Stasevich et al 2014) in 1% NGS in HMK for 1 h. After being washed with HMK four times, the cells were incubated with diluted secondary antibodies: goat antimouse IgG Alexa Fluor 488 (1:500, A11029; Thermo Fisher Scientific) in 1% NGS in HMK for 1 h followed by a four washes with HMK. DNA staining and mounting were performed as described above. Optical sectioning images were recorded with a 200 nm step size using a DeltaVision microscope (Applied Precision) as described in the section "Cell lines, DNA construction, and establishment of stable cell lines."

For RAD21 and RNAPII-Ser5P staining, the mean intensities of the nuclear signals after background subtraction (the signals outside nuclei) were calculated and plotted.

### Chromatin isolation, condensate imaging, and condensation assay by static light scattering

Fresh chicken blood was obtained from the wing vein of Tosa-jidori. Briefly, 1 ml of fresh chicken blood was lysed with 10 ml of MLB (60 mM KCl, 15 mM NaCl, 15 mM Hepes, pH 7.3, 2 mM MgCl$_2$, 0.1% NP-40, and 1 mM PMSF) for 10 min on ice. After centrifugation at 1,200$g$ at

4°C for 5 min, the supernatant was removed and resuspended in 10 ml of MLB. This step was repeated four times before the samples were ready for chromatin purification. Chromatin purification was carried out as described by Ura and Kaneda (2001), with some modifications. The nuclei (equivalent to ~2 mg of DNA) in nuclei isolation buffer (10 mM Tris–HCl, pH 7.5, 1.5 mM MgCl$_2$, 1.0 mM CaCl$_2$, 0.25 M sucrose, and 0.1 mM PMSF) were digested with 50 U of micrococcal nuclease (Worthington) at 30°C for 2 min. The reaction was stopped by adding ethylene glycol tetraacetic acid to a final concentration of 2 mM. After being washed with nuclei isolation buffer, the nuclei were lysed with lysis buffer (10 mM Tris–HCl, pH 8.0, 5 mM EDTA, and 0.1 mM PMSF) on ice for 5 min. The lysate was dialyzed against dialysis buffer (10 mM HEPES-NaOH, pH 7.5, 0.1 mM EDTA, and 0.1 mM PMSF) at 4°C overnight using Slide-A-Lyzer (66380; Thermo Fisher Scientific). The dialyzed lysate was centrifuged at 20,400$g$ at 4°C for 10 min. The supernatant was recovered and used as the purified chromatin fraction. The purity and integrity of the chromatin protein components were verified by 14% SDS–PAGE (Fig S3C). To examine average DNA length of the purified chromatin, DNA was isolated from the chromatin fraction and electrophoresed in 0.7% agarose gel (Fig S3D).

Samples of chicken chromatin (2 $\mu$g) were incubated with 0.5 or 2.5 mM of MgCl$_2$ for 15 min on ice and spun onto poly-L-lysine–coated coverslips by centrifugation at 2,380$g$ at 4°C for 15 min. The chromatin was gently fixed with 2% FA in the same buffer. After DNA staining (DAPI), the coverslips were sealed with nail polish. Optical sectioning images were recorded with a 200-nm step size using a DeltaVision microscope and decon-volved to remove out-of-focus information. Projected images from five sections were shown as described previously (Maeshima et al, 2016).

To analyze static light scattering by chicken chromatin, diluted chicken chromatin was centrifuged at 20,400$g$ for 1 min, then su-pernatant (200 $\mu$l) was used for analysis. Static light scattering at 90° angle was measured using a fluorescence spectrophotometer (F-4500; HITACHI) at a wavelength of 350 nm. A 10 mM solution of MgCl$_2$ was titrated into the samples containing indicated con-centrations of 1,6-HD to obtain the desired final Mg$^{2+}$ concentra-tions. The value measured at 0 mM was subtracted from all other measurements as background. After background subtraction, the resultant values were normalized to the peak value. Mean values from four experiments were plotted with their SDs.

## Data Availability

All data needed to evaluate the conclusions in the paper are present in the paper and/or the Supplementary Materials. Addi-tional data related to this paper may be requested from the authors.

## Supplementary Information

# Acknowledgements

We are grateful to Dr. MT Kanemaki for valuable help and for providing HCT116 cells expressing mClover-MED14, Dr. KM Marshall for critical reading and editing of this manuscript, Dr. H Kimura for his RNAPII antibody, Dr. S Hirose for critical reading of this manuscript, and Dr. T Hirose for helpful discussion. We thank Maeshima lab members for helpful discussions and support. Y Itoh was supported as a National Institute of Genetics Post-doctoral Fellow and is currently a Japan Society for the Promotion of Science (JSPS) Fellow. This work was supported by JSPS and MEXT KAKENHI grants (19K23735 and 20J00572 to Y Itoh; 18K06187 to S Ide; 19H05273 and 20H05936 to K Maeshima), a Japan Science and Technology Agency CREST grant (JPMJCR15G2 to K Maeshima), the Takeda Science Foundation (to K Maeshima), and the Uehara Memorial Foundation (to K Maeshima).

## Author Contributions

Y Itoh: conceptualization, funding acquisition, investigation, visualization, and writing—original draft, review, and editing.
S Iida: investigation, visualization, and writing—original draft, review, and editing.
S Tamura: investigation, visualization, and writing—original draft.
R Nagashima: conceptualization and investigation.
K Shiraki: writing—original draft.
T Goto: resources.
K Hibino: writing—review and editing.
S Ide: funding acquisition, investigation, and writing—original draft, review, and editing.
K Maeshima: conceptualization, supervision, funding acquisition, investigation, visualization, and writing—original draft, review, and editing.

## Conflict of Interest Statement

The authors declare that they have no conflict of interest.

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
