## [Reviewer comments · Life Science Alliance]

Life Science Alliance

1,6-hexanediol rapidly immobilizes and condenses chromatin in living human cells

Yuji Itoh, Shiori Iida, Sachiko Tamura, Ryosuke Nagashima, Kentaro Shiraki, Tatsuhiko Goto, Kayo Hibino, Satoru Ide, and Kazuhiro Maeshima

DOI: <https://doi.org/10.26508/lsa.202001005>

Corresponding author(s): Kazuhiro Maeshima, National Institute of Genetics

Review Timeline:	Submission Date:	2020-12-21
	Editorial Decision:	2020-12-21
	Revision Received:	2021-01-09
	Editorial Decision:	2021-01-13
	Revision Received:	2021-01-13
	Accepted:	2021-01-19

Scientific Editor: Shachi Bhatt

Transaction Report:

Please note that the manuscript was previously reviewed at another journal and the reports were taken into account in the decision-making process at Life Science Alliance.

December 21, 2020

Re: Life Science Alliance manuscript #LSA-2020-01005-T

Prof. Kazuhiro Maeshima
National Institute of Genetics
Chromosome Science
Yata 1111
Mishima, Shizuoka 411-8540
Japan

Dear Dr. Maeshima,

Thank you for transferring your manuscript entitled "1,6-hexanediol rapidly immobilizes and condenses chromatin in living human cells" to Life Science Alliance (LSA). The manuscript was assessed by expert reviewers.

For a brief overview, the manuscript was reviewed at one of the LSA alliance journals. The reviewers at the previous journal were appreciative of the quality of the data, but the paper was ultimately rejected given some concerns about scope. The manuscript and reviewers' comments were shared with LSA, with the authors' consent, and given the high quality of the data and the enthusiasm from the reviewers, the manuscript was deemed to be publishable at LSA with minor revisions, as follows:

- please address the minor text edits requested by Rev 1
- please provide a discussion (possibly some data, only if easily attainable) about the toxicity of 1,6 hexanediol (Rev 2, pt 1).

The revised manuscript does not need to be re-reviewed at LSA.

The typical timeframe for revisions is three months. Please note that papers are generally considered through only one revision cycle.

Thank you for this interesting contribution to Life Science Alliance. We are looking forward to receiving your revised manuscript.

Sincerely,

Shachi Bhatt, Ph.D.
Executive Editor
Life Science Alliance
<https://www.lsjournal.org/>
Tweet @SciBhatt @LSAJournal

- A letter addressing the reviewers' comments point by point.
- An editable version of the final text (.DOC or .DOCX) is needed for copyediting (no PDFs).
- High-resolution figure, supplementary figure and video files uploaded as individual files: See our detailed guidelines for preparing your production-ready images, <https://www.life-science-alliance.org/authors>
- Summary blurb (enter in submission system): A short text summarizing in a single sentence the study (max. 200 characters including spaces). This text is used in conjunction with the titles of papers, hence should be informative and complementary to the title and running title. It should describe the context and significance of the findings for a general readership; it should be written in the present tense and refer to the work in the third person. Author names should not be mentioned.

B. MANUSCRIPT ORGANIZATION AND FORMATTING:

Response to reviewers:

We would like to thank the reviewers for their thoughtful and insightful comments. We believe we have improved the quality of our manuscript by constructively responding to their suggestions and comments. Our point-by-point responses are provided below.

Reviewer#1

The authors present measurements of time-dependent nucleosome mobility (mean-squared displacement) in multiple human-derived cell lines following treatment with the aliphatic alcohol 1,6-hexanediol (hereafter 1,6-HD). The authors confirm previous reports that 1,6-HD disrupts nuclear condensates, including coilin-containing Cajal bodies and MED14-containing transcriptional condensates, in a dose dependent manner. The primary observation in this work is that the mobility of chromatin in live cells, for which nucleosomes are used as a proxy, is sharply attenuated by treatment with 1,6-HD. The authors convincingly demonstrate that the reduction in nucleosome mobility following 1,6-HD treatment i) increases in severity with increasing dosage, ii) persists for at least 90 minutes following wash-out at concentrations of 5 % and higher, and iii) is not unique to HeLa cells. To begin uncovering molecular mechanism underlying this observation, the authors demonstrate that reduction of nucleosome mobility by 1,6-HD requires neither the presence of the complete cohesion complex nor active RNA polymerase II in HeLa cells. Further, the authors show that treatment with the related aliphatic alcohol 2,5-hexanediol, which lacks the "droplet melting" activity of 1,6-HD, results in a comparable dose-dependent reduction in nucleosome mobility.

A second observation reported by the authors is that the spatial distribution of fluorescently-labeled nucleosomes in HeLa cells changes in a dose-dependent manner upon 1,6-HD treatment. The authors also find that the threshold concentration of MgCl₂ required for Mg²⁺-mediated clustering of purified chromatin, as detected by light scattering, is reduced by increasing concentrations of 1,6-HD.

The authors propose that attractive interactions between chromatin and aliphatic alcohols like 1,6-HD lead to preferential displacement of chromatin's solvating water by the alcohol and consequently an increased propensity for chromatin clustering

mediated by alcohols with multiple hydroxyl groups (hexanediols have two). The authors also propose that the 1,6-HD-dependent reduction in nucleosome mobility results from the change in clustering. Both of these propositions are plausible, though the authors do not test either of them.

The principle experimental methods used in this work were developed and published previously by a subset of the authors (1). The data in the current manuscript appear technically sound and, for those researchers using 1,6-HD, add to a growing list of caveats and "things to keep in mind" when interpreting data obtained with the compound (2-5).

Additional significant insight into the molecular mechanisms underlying any of 1,6-HD's three reported activities (condensate disruption, reduction in nucleosome mobility, and chromatin reorganization) would increase the impact of the present contribution to the fields of condensate and/or chromatin biology, as would testing the model proposed by the authors. My suspicion, however, is that major progress towards understanding the mechanism of action of 1,6-HD on droplets or chromatin is outside the scope of the current work, owing to the likely need for very different experimental approaches. In light of this, my remaining specific comments are all relatively minor.

We would very much appreciate Reviewer#1's insightful comments which were very useful for improving our manuscript.

Minor Issues and comments:

1) Language and discussion of LLPS

There are several sentences in the introduction and discussion that read to me as awkward, confusing, and sometimes wrong. Given the rapid development and interdisciplinary nature of the condensate biology field, this is somewhat understandable. Fortunately, it is easy to fix here.

· In first paragraph of the introduction, the sentence starting with "According to the LLPS principle..." reads awkwardly, and may inadvertently be missing a word or two. I suspect the intended sentiment is along the lines of "In principle, the organization of a cell's interior into distinct compartments facilitates the spatial and temporal regulation

of complex biochemical reactions." I would avoid phrasing like "the LLPS principle," as it is vague while giving the impression of specificity. The implications of LLPS in cells are not established at the level of a principle like "Le Chatelier's Principle" in chemistry, or the "Principle of Least Action" in physics. And even if the cellular implications were fantastically well-established, a principle based on them would require a name emphasizing the application to cells, since LLPS occurs in non-cellular contexts as well.

We agree with Reviewer#1 that the phrase “According to the LLPS principle...” might confuse the readers. In the revised manuscript, to avoid confusion, we deleted the phrase.

· The next sentence in the same paragraph, "How these dynamic biomolecular condensates/bodies form by LLPS remains unclear and the subject of debate for many cell biologists." is easy to misread and (I presume inadvertently) misleading. Since phase separation is a process by which molecular assemblies may form, a statement about how they form by phase separation is almost tautological. Another interpretation is that this sentence is about uncovering the molecular driving forces controlling the phase behavior of individual proteins and condensates. This is certainly an active area of research, but to my mind not a particularly contested one, certain disease-associated proteins like FUS notwithstanding. Whether any of the myriad biomolecular condensates in cells form by phase separation or another process is certainly a hotly debated topic. The papers cited, however, all represent the view that they do. McSwiggen et al., which the authors cite elsewhere, provides a critical evaluation of the evidence upon which claims of LLPS in cells are often based and advocates for the exploration of alternative interpretations. If this is debate to which the authors are referring, it would be appropriate to cite McSwiggen here.

We thank Reviewer#1 for his/her critical comments. As suggested, we rephrased the last sentence of the first paragraph as follow:

“whether all of these dynamic biomolecular condensates/bodies form by LLPS or some form by another process remains unclear and the subject of debate for many cell biologists (McSwiggen et al 2019)”

· *In the second paragraph of the introduction, I would recommend against the use of "LLPS interactions" for the same reasons I oppose "LLPS principle."*

We agree and to avoid confusion, we rephrased “LLPS interactions” to be “interactions”.

· *In the concluding sentence of introductory paragraph three, I would advocate for a bit more nuance in the author's statement that "the use of 1,6-HD to study protein/protein condensates/bodies is widely accepted." I agree that it is widely used. Stating "widely accepted" gives the impression that there is consensus approving its broad use. In contrast, several prominent authors have pointed out significant limitations and caveats to its use in the context of biomolecular condensates (2-5), a literature to which the current manuscript adds.*

We rephrased “widely accepted” into “widely used” and added the references that Reviewer#1 suggested into the Introduction:

“However, while 1,6-HD is widely used to study protein/protein condensates/bodies, some reports have pointed out significant limitations and caveats to its use in the context of biomolecular condensates/bodies (Alberti et al 2019, Kroschwald et al 2017, Lin et al 2016, McSwiggen et al 2019). Indeed the cellular effects of 1,6-HD, especially its effects on chromatin in living cells, remain unclear.”

· *In the third paragraph of the discussion, the authors write that "1,6-HD has been accepted as an additive to discriminate liquid droplets formed by LLPS in cells." I would again advocate for more nuance here, and suggest that 1,6-HD has been "identified" as compound which can be used to distinguish between liquid-like and solid-like structures in cells and in vitro, and cite Kroschwald et al. Importantly, the response of a structure to 1,6-HD is not informative with regard to the process by which the structure formed (5).*

We agree with Reviewer#1 that the description was confusing. As suggested, we

have rephrased the sentence as follows:

“1,6-HD has been identified as a compound that can be used to distinguish between liquid-like and solid-like structures in vitro and in living cells (Kroschwald et al 2017, Lin et al 2016)”

· In the subsequent sentence in the same paragraph, the authors state that "...this property of 1,6-HD works properly if the liquid droplets are composed only of proteins." This is incorrect. Many condensates that respond to 1,6-HD treatment, including the yeast P-bodies examined in Kroschwald et al, contain substantial amounts of RNA in addition to protein. Comments from the author in the discussion of whether and to what extent they expect that RNA might respond differently to 1,6-HD than DNA would be welcome.

We agree with Reviewer#1's comment: several condensates are also composed of proteins and RNAs. We rephrased the sentence as follows:

"...this property of 1,6-HD works properly if the liquid droplets are composed of proteins or possibly proteins bound with RNAs"

· In the last sentence of the 4th paragraph of the discussion, the authors write that "another mechanism is also possible." It is unclear whether the authors are referring to a specific alternative hypothesis, or merely that there could be alternatives in principle. If the authors have an alternative hypothesis, I encourage them to state it here.

To clarify this point, we rephrased the sentence as follows:

“Another mechanism of how 1,6-HD acts on chromatin in the cell is also possible,…”

2) Permanently and irreversibly frozen chromatin

The authors show that, at concentrations of 5% or higher, nucleosome mobility does not recover appreciably within 90 minutes following wash-out. This is not the same thing as the chromatin being irreversibly frozen. While it certainly could be that the reduction in mobility is effectively irreversible, that hasn't been shown. I would therefore advocate for softer language. If changes in e.g. cell viability following treatment with high

concentrations limit the ability to measure mobility beyond ~ 90 min, that would be useful to report.

Following Reviewer#1's suggestion, we removed the phrases "irreversibly" and "permanently" in the revised manuscript.

As addressed to Reviewer#2's comment below, we added the cell viability result after 1,6-HD treatment (Table 1).

3) Number of nucleosome trajectories per cell

The authors generally do an excellent job of annotating their figure captions. Thank you! One thing that is unclear to me, however, is the "average values of nucleosome trajectories used per cell." In Fig 2, for instance, the average is given as 1300-1800. This is WAY more points than the apparent number of individually discernable puncta in Fig. 2c. How are so many trajectories being obtained per cell? Is each displacement in within a trajectory being counted individually (e.g. counting a track 21-frames long as 20-displacements instead of 1 trajectory?)

Only a single frame from the single-nucleosome movie data was shown in Fig 2C. Each frame contained around 100 points and we analyzed more than 100 frames for each cell. Therefore the total trajectories per cell was often more than 1000.

4) Figure 5

There are a few small issues here. First, the microscopy images in Fig. 5A are a little difficult to interpret. A zoomed-in view of one or a few of the chromatin clusters would help. Second, in Fig. 5C, the placement of the molecules in the key just below the hydrophobicity color scale gives the impression that the molecules are being ranked by hydrophobicity, in which case it's problematic for 1,6-HD and water to have the same hydrophobicity, and for Mg²⁺ to be on the hydrophobic end of the scale. The meaning of the question mark in the right-hand panel is unclear. Is this to denote that the whole thing is a hypothesis? Or that the particular orientation of 1,6-HD is a unknown? Also, labeling the left- and right-hand schematics with titles like {plus minus} 1,6-HD would be helpful.

We agree with Reviewer#1. We magnified the images in Fig. 5A and improved Fig. 5C to avoid confusion.

Reviewer#2:

Since the discovery of liquid-liquid phase separation (LLPS) in cells, a number of studies have used 1,6-hexanediol to disrupt known or putative phase condensates in cells. These authors begin by confirming that 1,6-hexanediol can disrupt Cajal bodies and transcription-associated phase condensates in HeLa cells. They then go on to probe the effects of 1,6-hexanediol on chromatin in HeLa cells and three other human cell types as well as on chick erythrocyte chromatin in vitro. They report that a 5 minute treatment with 1,6-hexanediol causes chromatin to "freeze", ceasing thermal motions to an extent similar to that seen by methanol fixation. This "freezing" of chromatin is not reversible for concentrations of 1,6-hexanediol of 5% or above, whereas the effects seen with treatment of cells with 2.5% 1,6-hexanediol are reversed by washing out the alcohol. The authors show that knocking down cohesin or inhibiting RNAPII transcription cause an increase in chromatin mobility, and this is reversed by 1,6-hexanediol. Perhaps surprisingly, they find that 2,6-hexanediol, which is much less efficient at disrupting phase condensates, also causes an irreversible "freezing" of chromatin. Using PALM microscopy, they show that the cessation of motion is accompanied by a visible condensation of the chromatin in cells. And finally, they show that the effects can be at least partly mimicked in vitro, as addition of 1,6-hexanediol augments the compaction of chick erythrocyte chromatin observed after treatment with Mg²⁺. The authors claim that the use of 1,6-hexanediol to study the condensation of chromatin should be approached with caution, as effects observed may not be due to effects on LLPS.

The Maeshima lab have made excellent use of their oblique illumination system to study chromatin movement within living cells, and they are masters of this technology. Likewise the PALM studies reported here and the in vitro system seem to be well performed and are clearly described. Thus, I have no technical issues with this MS.

We would like to thank Reviewer#2 for his/her critical comments, which were very useful for improving our manuscript.

Nonetheless, I do not believe that this MS is suitable for publication in the journal for two reasons.

-1- Based on experiments done in my own lab and my conversations with others who work on LLPS, I have learned that 1,6-hexanediol is highly toxic to cells. We have found that even brief treatments with 5% or 10% rapidly kill cells. If the cells are dead, this would of course explain why the treatments with higher concentrations of the alcohol "freeze" the chromatin irreversibly. Thus, before the MS could be accepted as biologically relevant, the authors would need to perform a series of studies to show that the cells were still physiologically active during and after the treatments with these higher concentrations of 1,6-hexanediol.

We appreciate Reviewer#2 for raising this critical point. Following the suggestion, we examined the viability of the cells treated with 1,6-HD and included this result in the revised manuscript:

“Cell viability remained comparably high after cells were treated for 30 min with 2.5% or 5% 1,6-HD. However, the viability decreased to about 2% in cells following a 30 min treatment with 10% 1,6-HD (Table 1). These results suggest that our observation of chromatin freezing by 1,6-HD treatment is not a direct consequence of cell death, while the treatment has considerable cell toxicity.”

-2- I do not take a clear biological message away from this study. In order to merit a place in the journal, I would expect the story to yield some novel insight into a biological mechanism. In theory, this could be possible if the mechanism of chromatin compaction induced by 1,6-hexanediol and 2,6-hexanediol could be related to some sort of biological process - for example mitotic chromatin compaction. However, if the message is solely technical - that a treatment which many people working on LLPS already feel is too harsh to be reliably used in studies of living cells causes chromatin to "freeze" - then I do not feel that this is a journal story. It might be more suitable for a journal like the Journal of Biological Chemistry.

We agree with Reviewer#2 that our work is a bit out of scope in the previous journal. Our main

point is that we provided experimental evidence demonstrating that liquid droplet results obtained using 1,6-HD should be carefully interpreted or reconsidered when these droplets are associated with chromatin. It should be emphasized that while some reports have pointed out significant limitations and caveats to its use in the context of biomolecular condensates/bodies (Alberti et al 2019, Kroschwald et al 2017, Lin et al 2016, McSwiggen et al 2019), there are no actual experimental results showing 1,6-HD effects on chromatin in living cells.

To clarify the significance of our paper, we improved the Abstract and Introduction parts.

Reviewer #3:

This manuscript addresses the effect of 1,6-HD on chromatin mobility and compaction. This is incredibly rare for me, but I had not a single issue with the work presented. I found it convincing, well presented, and quite clear-cut.

We would very much appreciate Reviewer#3's encouraging comment on our work.

What was not clear to me was the final take-home message and the readers for whom this take-home message would be important. People seem to have increasingly turned to the application in vivo of 1,6-HD to apply a test to determine whether their favorite cell body is a "LLPS". There have been review articles, such as by Tjian and colleagues, who have raised concerns about this approach. As shown in this manuscript, 1,6-HD can cause other effects that could directly or indirectly feedback and change some aspect of cell physiology, including the appearance of a "body", through a separate phenomenon than LLPS. Therefore, if one added 1,6-HD to cells and saw a disappearance of a body, then this would only be consistent with the possibility of LLPS but would not disprove alternative mechanisms for the formation of whatever they were studying other than LLPS. But that seems to be exactly where the field already stands without this new information. I suppose one might argue that it adds an example where something other than LLPS is happening as an effect of adding 1,6-HD to live cells.

Our main point is that liquid droplet results obtained using 1,6-HD should be

carefully interpreted or reconsidered when these droplets are associated with chromatin. As addressed to Reviewer#2's comment above, it should be again emphasized that while some reports have pointed out significant limitations and caveats to its use in the context of biomolecular condensates/bodies (Alberti et al 2019, Kroschwald et al 2017, Lin et al 2016, McSwiggen et al 2019), there are no actual experimental results showing 1,6-HD effects on chromatin in living cells.

Incidentally, I thought that the authors' nice demonstration that another alcohol had the same effect on chromatin mobility but without affecting LLPS should have been emphasized more in the Discussion to emphasize the separate from LLPS and potentially indirect effects that can be induced by 1,6-HD.

We would like to thank Reviewer #3 for his/her useful comment to improve our manuscript and added the following sentences in Discussion:

“Interestingly, another aliphatic alcohol, 2,5-HD, which has a much lower melting activity of droplets formed by LLPS (Lin et al 2016), had a comparable motion suppression effect to 1,6-HD (Fig. 3D). This finding indicates that the observed 1,6-HD ‘freezing’ action on chromatin organization is distinct from its disruption activity of liquid droplets formed by LLPS.”

But getting back to the message of the take-home message, I don't imagine that most people studying chromatin mobility would suddenly decide to use 1,6-HD in their experiments. Conversely, people studying something else other than chromatin would not stop using 1,6-HD after reading about its effects on chromatin mobility to test whether they could provide some additional support for their focus of study being caused by LLPS.

Anyone studying the possible role of LLPS on chromatin organization and seeing a decrease in apparent LLPS through loss of some condensed DNA structure using 1,6-HD would still probably consider the results informative, given that it runs opposite the general trend of increasing condensation of chromatin.

So it would seem that the take-home message of this manuscript would be important only to that subset of investigators that saw some type of condensed chromatin that they suspected might be caused by LLPS, treated cells with 1,6-HD and saw that the condensed chromatin structure they were studying remained, and might then have concluded that the phenomenon they were studying was NOT LLPS if they had not seen the results of this manuscript. If the chromatin phenomenon really was driven by LLPS AND if it resembled sufficiently the condensed state that 1,6-HD causes to bulk chromatin, then I guess there would be the possibility that the actual chromatin LLPS was disrupted, but then the chromatin was condensed by the effect of 1,6-HD in just the right way as to look like the original structure was not perturbed.

We disagree with Reviewer#3's point because he/she may have only paid attention to the chromatin condensation effect by 1,6-HD. However, since our single-molecule imaging detected change(s) of local chromatin environments in the 1,6-HD-treated cells, it is very likely that 1,6-HD affected behaviors of various chromatin binding proteins. Our study thus suggests that 1,6-HD-sensitivity cannot be evidence for proving that cellular condensates/bodies of protein/DNA complexes, including chromatin, are formed by LLPS. To clarify the significance of our paper, we have improved the Abstract and Introduction parts.

January 13, 2021

RE: Life Science Alliance Manuscript #LSA-2020-01005-TR

Prof. Kazuhiro Maeshima
National Institute of Genetics
Chromosome Science
Yata 1111
Mishima, Shizuoka 411-8540
Japan

Dear Dr. Maeshima,

Thank you for submitting your revised manuscript entitled "1,6-hexanediol rapidly immobilizes and condenses chromatin in living human cells". We would be happy to publish your paper in Life Science Alliance pending final revisions necessary to meet our formatting guidelines.

Along with the points listed below, please also attend to the following,

- please extract the video files from the .Zip folder and upload them separately
- please make sure the manuscript sections are aligned in accordance to LSA's formatting guidelines: please separate the Figure legends and Supplemental Figure legends into separate sections

A. FINAL FILES:

-- Summary blurb (enter in submission system): A short text summarizing in a single sentence the study (max. 200 characters including spaces). This text is used in conjunction with the titles of papers, hence should be informative and complementary to the title. It should describe the context

and significance of the findings for a general readership; it should be written in the present tense and refer to the work in the third person. Author names should not be mentioned.

B. MANUSCRIPT ORGANIZATION AND FORMATTING:

Sincerely,

Shachi Bhatt, Ph.D.
Executive Editor
Life Science Alliance
<https://www.lsjournal.org/>
Tweet @SciBhatt @LSAJournal

January 19, 2021

RE: Life Science Alliance Manuscript #LSA-2020-01005-TRR

Prof. Kazuhiro Maeshima
National Institute of Genetics
Chromosome Science
Yata 1111
Mishima, Shizuoka 411-8540
Japan

Dear Dr. Maeshima,

Thank you for submitting your Research Article entitled "1,6-hexanediol rapidly immobilizes and condenses chromatin in living human cells". It is a pleasure to let you know that your manuscript is now accepted for publication in Life Science Alliance. Congratulations on this interesting work.

DISTRIBUTION OF MATERIALS:

Again, congratulations on a very nice paper. I hope you found the review process to be constructive and are pleased with how the manuscript was handled editorially. We look forward to future exciting submissions from your lab.

Sincerely,

Shachi Bhatt, Ph.D.

Executive Editor

Life Science Alliance

<https://www.lsjournal.org/>
